# Mapping the Unmet Informational Needs of Young Portuguese Female Cancer Survivors: Psychometric Validation of a Multidimensional Scale

**DOI:** 10.3390/healthcare13141757

**Published:** 2025-07-20

**Authors:** Luana Almeida, Ana Bártolo, Sara Monteiro, Isabel S. Silva, Ana Conde, Alexandra M. Araújo, Luiz Lourenço, Isabel M. Santos

**Affiliations:** 1Department of Psychology and Education, Portucalense University, 4200-072 Porto, Portugal; luana_monteiro_almeida@hotmail.com (L.A.); ana.bartolo@upt.pt (A.B.); anac@upt.pt (A.C.); amaraujo@upt.pt (A.M.A.); 2CINTESIS.UPT@RISE-Health, Portucalense University, 4200-072 Porto, Portugal; 3CINTESIS@RISE, Department of Education and Psychology, University of Aveiro, 3810-193 Aveiro, Portugal; smonteiro@ua.pt (S.M.); luizkung@gmail.com (L.L.); 4Department of Social Sciences and Management, Open University, 1269-001 Lisboa, Portugal; 5Center for Global Studies, Open University, 1269-001 Lisboa, Portugal; 6INSIGHT: Piaget Research Center for Human and Ecological Development, Piaget Institute—ISEIT/Viseu, 3515-776 Viseu, Portugal; isabel.silva@ipiaget.pt; 7Centre for Functional Ecology—Science for People & the Planet (CFE), TERRA Associated Laboratory, University of Coimbra, 3000-456 Coimbra, Portugal; 8William James Center for Research, Department of Education and Psychology, University of Aveiro, 3810-193 Aveiro, Portugal

**Keywords:** information needs, oncology, validity, reliability, SIPYF-CPS

## Abstract

**Background/Objectives:** Young female cancer survivors often face specific informational needs related to the physical and emotional effects of cancer and its impact on life plans, particularly fertility and parenthood. However, few tools are tailored to assess these needs during this critical life stage. This study aimed to (i) validate a multidimensional measure—the Satisfaction with Information Provided to Young Oncology Patients Scale (SIPYF-CPS)—to assess the specific informational needs of young adult female cancer survivors; and (ii) explore preferences regarding the provision of information and counseling. **Methods:** A total of 124 women (M[age] = 38.18; SD = 5.49; range 21–45), 76.6% diagnosed with breast cancer, participated in the study. Psychometric analyses included exploratory factor analysis and correlation coefficients to assess reliability and construct validity. Convergent validity was evaluated through standardized measures of anxiety, reproductive concerns, and quality of life. **Results:** A final 22-item measure demonstrated strong reliability and validity, capturing four factors: (i) Disease-Related Information, (ii) Symptoms and Functional Limitations, (iii) Implications for Fertility and Parenthood, and (iv) Support Services. Participants expressed low satisfaction with information on fertility preservation, sexual health, and support services. Lower satisfaction was moderately associated with higher anxiety and depression while positively related to quality of life. Most participants preferred phased, face-to-face communication throughout the illness trajectory. **Conclusions:** The SIPYF-CPS is a valid, multidimensional tool that captures the complex and evolving informational needs of young female cancer survivors. Its clinical use may promote earlier, personalized, and emotionally responsive communication—supporting psychological well-being, informed decision-making, and long-term survivorship care.

## 1. Introduction

Oncological disease represents a significant challenge to global public health, with both short- and long-term impacts on individuals’ functionality and quality of life. This condition can arise at any stage of human development and has shown a consistent increase in global incidence. Among the various types of cancer, breast cancer remains the most prevalent among women, with approximately 2,296,840 new cases reported in 2022. It is also the most diagnosed cancer among women of reproductive age (20–39 years), with an incidence rate of 19.7 per 100,000, followed by cervical cancer [1]. In this context, growing attention has been directed toward understanding the implications of an oncological diagnosis in young women, who face unique and complex challenges that intersect with key developmental milestones and life transitions [2]. These challenges—spanning physical, emotional, cognitive, and social domains—extend throughout the continuum of cancer care, from diagnosis to survivorship.

Among the most distressing aspects of a cancer diagnosis in young women are concerns related to fertility and reproductive planning, especially considering the potential risk of infertility [2,3]. Research has highlighted a range of concerns in this population, including the preservation of fertility, future family planning, and the transmission of genetic risks to offspring [4,5]. These issues can significantly affect women’s long-term life goals and psychological well-being.

Certain cancer treatments—such as cytotoxic chemotherapy, pelvic or abdominal radiation, and cranial irradiation—are known to compromise reproductive potential [6,7,8]. Chemotherapy regimens containing alkylating agents are particularly gonadotoxic, as they target all phases of the cell cycle and can irreversibly damage both proliferating and dormant cells, including primordial follicles that constitute the ovarian reserve [9]. Agents such as cyclophosphamide, procarbazine, and busulfan are especially harmful, with the potential to cause permanent ovarian failure, infertility, and early menopause [10,11]. In addition, adjuvant hormone therapy—frequently prescribed in hormone receptor-positive breast cancer—has increasingly been recommended for extended durations (typically 7 to 10 years [12]). Although not directly gonadotoxic, these therapies have teratogenic effects, necessitating the postponement of pregnancy, which can contribute to natural ovarian aging and further compromise fertility [9]. Together, these findings underscore the multifaceted and deeply personal impact that a cancer diagnosis and its treatment can have on young women. Accordingly, it is essential to address age-specific needs—particularly those related to access to accurate information and psychosocial support—in order to promote informed decision-making and uphold these individuals’ autonomy in planning for their future.

### 1.1. Understanding Information Needs in the Context of a Cancer Diagnosis

Meeting informational needs plays a central role in how patients adapt to a cancer diagnosis. In the healthcare context, information needs are generally understood as gaps or deficits in knowledge. More specifically, drawing on Ormandy’s [13] theoretical framework, “*a patient’s information need is defined as the recognition that his or her knowledge is inadequate to satisfy a goal, within the context/situation in which he or she finds himself or herself at a specific moment in time*” (p. 92).

An increasing number of studies have sought to identify the main informational needs experienced by patients, especially given the burden of a cancer diagnosis and its treatment. The findings suggest that these needs tend to cluster into categories such as psychological, health system and information, physical and daily living, care and support, and sexual health, e.g., [14,15,16]. Kav et al. [17], for example, found that 42.9% of patients rated the information that they received as “partially sufficient,” identifying major gaps regarding chemotherapy, treatment options, and side effects. Likewise, the literature has emphasized that treatment-related side effects—especially those related to chemotherapy [18]—as well as health system information and psychological concerns (e.g., fear of cancer recurrence and lack of information about cancer control) are among the most frequently reported unmet needs. There is well-established evidence that unmet informational needs are closely associated with greater functional impairments [15].

Research has also highlighted a marked lack of information regarding reproductive and sexual health following cancer and its treatment—particularly among women [19,20]. Reproductive health, in particular, has become an area of growing concern in the literature, especially for young women, who often report distress and uncertainty regarding this topic [21]. Although many of these women may retain their reproductive potential after treatment, the risk of premature ovarian failure—such as early menopause—remains substantial [10]. These concerns frequently influence decisions about whether to continue or interrupt treatment, pursue fertility preservation, or reassess family planning goals [22].

Nevertheless, studies show that many young cancer survivors do not receive fertility-related counseling as recommended by clinical guidelines [10,23,24]. Even when counseling is offered, it often fails to fully address patients’ concerns [25]. This lack of knowledge regarding reproductive health after treatment, combined with uncertainty around fertility outcomes, may have long-term consequences—including reduced quality of life and poorer mental health outcomes [10,22]. Patients with lower quality of life often report more unmet informational needs and higher levels of anxiety about fertility and the ability to carry a pregnancy to term. In fact, 45% of these patients expressed a desire to have children within three years of diagnosis [22].

Given these realities, it is essential to assess unmet informational needs in young women from a multidimensional perspective that considers their life stage and developmental context. Several international instruments have been developed to evaluate informational needs in oncology, such as the Information and Support Needs Questionnaire (ISNQ [26]) and the Cassileth’s Information Styles Questionnaire [27]. However, these tools are not specifically designed for women of reproductive age and do not incorporate key dimensions of informational need as defined by Ormandy [13]—namely, the goal, context/situation, and specific point in time at which the information is required.

In Portugal, instruments such as the Short-Form Survivor Unmet Needs Survey [28] and the Supportive Care Needs Survey—Short Form 34 (SCNS-SF34 [29]) have been translated and validated. Although these tools assess a broad range of needs (e.g., psychological, informational, supportive, sexual), they do not specifically address fertility-related issues or parenting goals. To bridge this gap, Silva et al. [30,31] initiated the development of a new assessment instrument: the Satisfaction with Information Provided to Young Female Cancer Patients Scale (SIPYF-CPS). Grounded in Ormandy’s theoretical assumptions regarding informational needs [13], this scale was designed to assess a broad spectrum of information-related demands across the cancer continuum—from diagnosis through survivorship—thereby opening new avenues for research on its psychometric properties.

### 1.2. Objectives and Hypotheses

The present study aimed to (i) explore the psychometric properties (reliability and construct validity) of the SIPYF-CPS; (ii) identify the main unmet needs of reproductive-age Portuguese female cancer survivors; and (iii) characterize their preferences regarding the delivery of information and counseling.

To evaluate the construct validity of the SIPYF-CPS, and in line with COSMIN recommendations [32], specific hypotheses regarding convergent validity were formulated, grounded in both theoretical frameworks and previous empirical findings. Prior research has consistently demonstrated that unmet informational needs are associated with elevated psychological distress, including symptoms of anxiety and depression (e.g., [33]). Therefore, satisfaction with informational needs, as measured by the SIPYF-CPS, was expected to exhibit moderate to strong negative correlations with anxiety and depression, indicating that lower satisfaction corresponds to greater psychological distress.

In addition, concerns related to fertility and reproductive health have been identified as significant stressors among young female cancer survivors, often intensified by inadequate information provision [10]. Accordingly, weak to moderate negative correlations between informational satisfaction and reproductive concerns were anticipated. Furthermore, domains of quality of life—particularly emotional, cognitive, and physical functioning—have shown positive associations with greater information satisfaction in oncology populations [15,22,34]. Based on this, moderate positive correlations between SIPYF-CPS scores and these quality-of-life domains were hypothesized, with weaker associations expected for social and role functioning. Together, these hypotheses provided a framework for evaluating the convergent validity of the SIPYF-CPS and guided the interpretation of the correlation analyses conducted in this study.

Ultimately, the results of this study are expected to contribute to the validation of a context-specific tool for assessing informational needs and to improve the understanding of unmet needs during cancer diagnosis and treatment. This knowledge may support the development of tailored interventions that are aligned with the preferences of young female cancer survivors regarding information and counseling delivery.

## 2. Materials and Methods

### 2.1. Study Design and Participants

The present study is a validation, observational, cross-sectional study that included a convenience sample. The inclusion criteria for participation were (i) being female; (ii) aged between 18 and 45 years at the time of study enrolment; (iii) having been diagnosed with cancer at least 12 months prior to participation; and (iv) being a native Portuguese speaker.

### 2.2. Measures

#### 2.2.1. Sociodemographic and Clinical Questionnaire

This questionnaire consisted of three sections: the first section gathered sociodemographic information, including questions about age, nationality, education level, marital status, and employment status. The second section focused on the participant’s clinical situation, including questions related to the type of cancer, timing of diagnosis, cancer stage at diagnosis, and other relevant aspects. The third section addressed reproductive health and parenting-related issues.

#### 2.2.2. Satisfaction with the Information Provided to Young Female Cancer Patients Scale (SIPYF-CPS)

The SIPYF-CPS is a self-administered scale developed by a research team as part of a project aimed at assessing satisfaction with the information provided during cancer diagnosis and care in young female cancer survivors [30,31]. The development process, illustrated in Figure 1, followed established best practices for scale construction and validation in health and behavioral research [35]. Before defining the theoretical domains and items, the authors conducted a comprehensive literature review to guide the measure’s development. Grounded in a patient-centered approach and informed by Ormandy’s [13] theoretical model of informational needs, the scale was structured around four overarching dimensions: (1) goals/purpose, (2) context, (3) situation, and (4) time.

Within this framework, the SIPYF-CPS was specifically designed to assess informational needs across four key content domains. The first is the Disease Information domain, which encompasses information on the type and stage of cancer, treatment options, symptoms, prognosis, and guidance on surveillance and risk-reducing behaviors—crucial for responding to the immediate demands of the disease. The second, the Functional Limitations domain, includes information related to restrictions on daily activities during and after treatment, work capacity, and engagement in leisure activities, depending on each patient’s clinical condition. The third, the Fertility and Parenting Implications domain, addresses concerns particularly relevant to women of reproductive age, such as the risk of infertility, early menopause, genetic risks to offspring, fertility preservation strategies, and alternatives for family building. The fourth, the Services and Organizations Information domain, focuses on internal motivations and concerns regarding access to services and support networks. The initial version of the SIPYF-CPS comprised 23 items distributed across these four domains, assessed on a 5-point Likert scale, ranging from “not at all satisfied” to “very satisfied.”

A core assumption of the scale’s underlying model is that informational needs are dynamic and vary over time. Accordingly, the SIPYF-CPS incorporates six additional closed-ended items that explore participants’ preferences concerning the timing and delivery format of information and counseling. To ensure content validity, the scale underwent independent review by a panel of psycho-oncology specialists. Additionally, the measure was pretested with a small group of cancer patients (n = 3), though this sample was not included in the main validation study. In the original study [30], preliminary psychometric analysis using a sample of 39 participants demonstrated high internal consistency, with a Cronbach’s alpha of 0.94 for the total scale. In this study, participants were also invited to provide feedback on the measure, further strengthening its content validity. Lower scores on the SIPYF-CPS indicate greater unmet informational needs.

#### 2.2.3. Hospital Anxiety and Depression Scale (HADS)

Validated for the Portuguese population by Pais-Ribeiro et al. [36], this self-report scale consists of 14 items and evaluates symptoms of depression (7 items) and anxiety (7 items) in two subscales. Items are scored using a Likert-type scale ranging from 0 to 4 points, with the total score for each subscale varying from 0 to 21. Anxiety and depression levels are classified as: “normal” (0–7), “mild” (8–10), “moderate” (11–14), and “severe” (15–21). The scale showed good internal consistency for the present sample, with Cronbach’s alpha values of 0.90 for the anxiety subscale and 0.86 for the depression subscale.

#### 2.2.4. Reproductive Concerns After Cancer Scale (RCACS)

The Reproductive Concerns After Cancer Scale (RCACS), developed by Gorman et al. [37] and validated for the Portuguese population by Bártolo et al. [38], is a self-report instrument comprising 18 items designed to assess fertility and parenthood concerns among young female cancer survivors. The Portuguese version evaluates reproductive concerns across five dimensions: fertility potential, children’s health risks and future life, partner disclosure, acceptance, and achieving pregnancy/having children. Participants are asked to indicate their level of agreement, with each statement using a 5-point Likert scale ranging from 1 (strongly disagree) to 5 (strongly agree). The total score for each dimension can range from 18 to 90 points, with higher scores indicating greater levels of concern. In the present study, the measure demonstrated good internal consistency within the sample, with a Cronbach’s alpha of 0.836.

#### 2.2.5. European Organization for Research and Treatment of Cancer Quality of Life Questionnaire Core30 (QLQ-C30)

This self-report questionnaire, developed by the Cancer Study Group on Quality of Life of the European Organization for Research and Treatment of Cancer (EORTC) and validated for the Portuguese population by Pais-Ribeiro et al. [39], assesses the quality of life in cancer patients. It consists of 30 items, 24 of which are grouped into nine subscales: five functional subscales (physical functioning, role functioning, emotional functioning, cognitive functioning, and social functioning), three symptom subscales (fatigue, nausea and vomiting, and pain), one global health/quality of life subscale, and six individual items assessing symptoms. Most items are responded on a 4-point Likert scale ranging from 1 (“Not at all”) to 4 (“Very much”), while the global health/quality of life subscale uses a scale that ranges from 1 (“Very Poor”) to 7 (“Excellent”) [40]. After a linear transformation of the raw score, the total score ranges from 0 to 100. Regarding the functional and global health perception scales, higher scores indicate better levels of functioning, whereas on the symptom scales, higher scores indicate worse outcomes in terms of symptoms and difficulties [40]. In the present study, only the results from the five functioning subscales were analyzed. These subscales demonstrated good internal consistency, with Cronbach’s alpha coefficients ranging from 0.730 to 0.883 for cognitive and social functioning, respectively.

### 2.3. Ethical Considerations and Recruitment Process

The present study was approved by an Independent Ethics Committee—the Ethics Committee of Instituto Piaget (CEIP)—as part of the project “SIDEbySIDE”—Young cancer survivors and their partners side by side: a dyadic approach to fertility-related distress. (Reference: P32–S52–10/05/2023). The questionnaire was located on the Limesurvey platform, hosted on the servers of Universidade Portucalense, and participants were recruited online. The study was disseminated through cancer survivors’ associations in the community, as well as through pages and groups on various social media platforms (e.g., Facebook and Instagram). All participants had access to information about the study prior to their participation, and to proceed, they needed to provide informed consent by selecting a checkbox. The study followed all ethical research procedures outlined in the Code of Ethics of Portuguese Psychologists and the Helsinki Declaration, ensuring the confidentiality of the participants.

### 2.4. Data Analysis

Descriptive analyses were performed using IBM SPSS Statistics (version 29). The properties of the SIPYF-CPS items were assessed by analyzing item frequencies, quartiles, kurtosis, skewness, and internal consistency (Cronbach’s α). The descriptive analysis also identified the main unmet informational needs of the participants and their preferences. For the study of the psychometric properties of the SIPYF-CPS, particularly construct validity, an exploratory factor analysis (EFA) was performed using Mplus software, version 6.12. Considering the ordinal nature of the questionnaire items, the weighted least squares means and variance adjusted (WLSMV) estimator was used. This method enabled the comparison of different factor solutions. To allow for inter-factor correlations, an oblique rotation (Oblimin) was employed. Only items with factor loadings greater than 0.40 were included in the final version of the measure [41,42]. Cronbach’s alpha (α) and McDonald’s Omega (ω) were used to assess the reliability of the final version of the SIPYF-CPS, specifically considering the dimensions identified in the factor analysis. Further evidence of construct validity, particularly convergent validity, was provided by testing associations between scale scores and theoretically related constructs such as anxiety, depression, reproductive concerns, and quality of life, using Pearson’s correlation coefficient. These analyses provided key insights into the relationship between unmet informational needs and the psychosocial adjustment of the participants.

## 3. Results

### 3.1. Sociodemographic and Clinical Characteristics of the Sample

The sample included 124 female cancer survivors aged between 21 and 45 years (M = 38.18 years; SD = 5.49). Geographically, participants were from the north (29.8%), center (23.4%), and south (43.5%) of mainland Portugal, with only 1.6% residing in the autonomous regions. Most participants had completed a college degree (65.3%), were married/cohabiting (68.5%), and were employed full-time—either self-employed or working for others (58.9%). The most frequently reported diagnosis was breast cancer (76.6%), with the average age at diagnosis being 35.77 years (SD = 5.67). Approximately 64.5% of participants had undergone surgery, 80.6% had received chemotherapy, and 54% had received radiotherapy. However, 65.3% were still undergoing treatment, including chemotherapy (20.2%) and hormonal therapy (38.7%). About 25.8% of the participants had comorbidities with other chronic diseases (e.g., asthma, hypertension, diabetes). At the time of participation, 46% of the young women were receiving psychological and/or psychiatric support. Regarding their reproductive history, about 58.9% had one or more biological children. About 41.1% were experiencing temporary amenorrhea, and 33.1% reported being in menopause. During clinical follow-up, 28.2% of participants reported being referred for medically assisted procreation consultation. Only 22.6% had undergone fertility care, such as embryo, oocyte, or ovarian tissue cryopreservation (18.5%) or other techniques such as oophoropexy (4.1%). Among participants, 33.1% reported that having a child (or another biological child) was important to them at the time of participation. Finally, it is important to highlight that most participants received their cancer-related care in a public hospital setting (72.6%). The sample characteristics are summarized in Table 1.

### 3.2. Unmet Informational Needs: Descriptive Analysis of SIPYF-CPS Items

The results indicate that a significant number of participants expressed dissatisfaction (responses of ‘not at all’ or ‘slightly’ satisfied) with the information provided about non-conventional complementary treatments (e.g., Reiki, acupuncture; 58.1%) in the context of cancer, as well as with the availability and functioning of informal support groups (52.4%). Additionally, there was a gap in the information provided regarding services offered by healthcare institutions, including psychological support, social services, and gynecological care, with 37.9% of participants reporting dissatisfaction. Unmet informational needs were also identified in areas related to sexual and reproductive health. For example, 49.2% of participants were slightly or not at all satisfied with the information about the implications of the disease on sexual life, while 46.8% expressed dissatisfaction with information on alternatives for building/continuing a family (e.g., adoption). Furthermore, 37.9% of participants felt inadequately informed about fertility preservation options. Over 30% of participants also reported dissatisfaction with information regarding physical treatment side effects (37.1%), potentially risky behaviors in the course of the disease (35.5%), expected emotional symptoms during treatment (33.8%), and limitations in work capacity and/or returning to work (32.3%) (see Table 2).

### 3.3. Psychometric Properties Assessment of the SIPYF-CPS

To test the psychometric properties of the SIPYF-CPS, which was used to assess satisfaction with the information provided to young women diagnosed with cancer, we explored its construct validity and reliability.

#### 3.3.1. Item Properties

All possible response options on the Likert scale were utilized for each item. The median of responses for most items was close to 3 (see Table 2). As shown in Table 3, no deviations from normality were observed, based on the absolute values of skewness (Sk < 3.0) and kurtosis (Ku < 7.0; [43]). All items demonstrated relatively large and positive corrected item–total correlations (r ≥ 0.31). Minimal variation in reliability was observed when individual items were excluded. Inter-item correlations were mostly above 0.30 (ranging from 0.10 to 0.81). Item 17 showed the lowest correlations with the other items.

#### 3.3.2. Exploratory Factorial Analysis

The sample size met the recommended criterion of at least five participants per item, ensuring appropriate conditions for conducting an exploratory factor analysis (EFA) [44]. The analysis was conducted using the WLSMV estimator, which is suitable for ordinal data. An oblimin rotation was applied to account for potential correlations among latent factors. Models specifying one to four factors were estimated and compared based on statistical fit indices, conceptual interpretability, and the structure of factor loadings.

The initial analysis used the original 23-item version of the scale. Fit indices showed progressive improvement with the inclusion of additional factors (see Table 4), with the four-factor solution yielding the best overall fit. Although some items showed standardized loadings close to the cutoff point of 0.40 on secondary factors, suggestive of cross-loading—for example, items 10 (“physical side effects of treatments”) and 14 (“services offered by the healthcare provider unit”)—these items demonstrated good item–total correlations in previous analyses, indicating consistent internal reliability. Moreover, removing these items worsened the overall model fit according to fit indices, reinforcing their contribution to the factor structure. Including these items in their conceptually related primary factor also improved the internal reliability of that specific factor, strengthening the theoretical justification for their retention in the scale. Only item 22 (“genetic risk for offspring”) was excluded because it did not load significantly on any factor (i.e., standardized loadings < 0.40; see Table 5) and exhibited a high residual variance (0.536), indicating poor fit and variance not accounted for by the factor structure. As a result, a new EFA was conducted using the remaining 22 items. The revised four-factor model demonstrated an improved model fit and a clearer factorial structure. Fit indices indicated acceptable fit: χ^2^(149) = 279.56, *p* < 0.001; χ^2^/df = 1.88; CFI = 0.97; RMSEA = 0.08, 90% CI [0.07–0.10]; and SRMR = 0.04. Although the chi-square test remained statistically significant—an expected outcome in large samples—other indices supported a well-fitting model, with an excellent comparative fit (CFI ≥ 0.95), low residual-based misfit (SRMR < 0.05), and acceptable approximate fit (RMSEA ≤ 0.10).

The final four-factor structure was theoretically meaningful and internally consistent. The first factor (F1), Disease-Related Information, comprised items 1 to 3, 6, 18, and 19 (6 items). The second factor (F2), Symptoms and Functional Limitations, included items 4, 5, 10 to 13, 20, and 21 (8 items). The third factor (F3), Implications for Fertility and Parenthood, encompassed items 7 to 9 and 23 (4 items), while the fourth factor (F4), Support Services, grouped items 14 to 17 (4 items; see Section A.1). Internal consistency analyses indicated satisfactory to excellent reliability, with Cronbach’s alpha coefficients ranging from 0.738 (Support Services) to 0.917 (Symptoms and Functional Limitations). Comparable values were observed for McDonald’s omega coefficients, further supporting the reliability of the instrument. Additionally, the total 22-item scale demonstrated excellent internal consistency, with both Cronbach’s alpha and McDonald’s omega yielding a coefficient of 0.94 (see Table 5). The four factors demonstrated moderate intercorrelations (ranging from 0.37 to 0.60), indicating that while related, they represent distinct dimensions within the construct.

#### 3.3.3. Exploring the Relationship Between Informational Needs, Distress, Reproductive Concerns, and Quality of Life as Evidence of Convergent Validity

Descriptive results revealed that the domain of Support Services (F4) had the lowest mean score, indicating it was perceived as the area with the most unmet needs. Although the domain of Implications for Fertility and Parenthood (F3) showed a relatively high mean, the high standard deviation (SD = 1.10) suggests notable variability in perceptions of need fulfillment (see Table 6).

To better understand the relationships among the identified domains of psychosocial needs, bivariate correlation analyses were conducted. These analyses indicated that satisfaction with informational needs was negatively associated with symptoms of anxiety (*r* = –0.464, *p* < 0.001) and depression (*r* = –0.500, *p* < 0.001). These findings suggest that lower satisfaction with the information provided was associated with higher levels of psychological distress among participants. Small to moderate significant correlations were found between all dimensions of the SIPYF-CPS and symptoms of anxiety (–0.261 ≤ *r* ≤ –0.488, *p* < 0.05) and depression (–0.309 ≤ *r* ≤ –0.495, *p* < 0.01). The weakest associations were observed for the Support Services dimension, while the strongest were found for the Symptoms and Functional Limitations dimension. A significant yet weak negative correlation was also found between overall satisfaction with informational needs and reproductive concerns reported by the young women in the study (*r* = –0.294, *p* = 0.001). A moderate negative association was specifically observed between reproductive concerns and the Disease-Related Information dimension (*r* = –0.320, *p* < 0.001). With regard to quality of life, moderate positive correlations were found between overall satisfaction with informational needs and emotional functioning (*r* = 0.397, *p* < 0.001), physical functioning (*r* = 0.338, *p* < 0.001), and cognitive functioning (*r* = 0.309, *p* < 0.001). Weaker but significant associations were found with social (*r* = 0.267, *p* = 0.006) and role functioning (*r* = 0.271, *p* = 0.005).

At the level of the SIPYF-CPS dimensions, small to moderate significant correlations were found between all quality-of-life functioning domains and the Symptoms and Functional Limitations dimension of the measure (0.287 ≤ *r* ≤ 0.411, *p* < 0.01) for social and emotional functioning, respectively). Regarding Disease-Related Information, moderate associations were also found with most quality-of-life domains, except for social (*r* = 0.210, *p* = 0.031) and role (*r* = 0.219, *p* = 0.024) functioning, which showed weaker but still significant correlations. Additionally, weak but statistically significant positive correlations were found between satisfaction with information related to Implications for Fertility and Parenthood and emotional (*r* = 0.202, *p* = 0.038), physical (*r* = 0.203, *p* = 0.037), and social (*r* = 0.251, *p* = 0.010) functioning. Finally, concerning satisfaction with Support Services, a small positive correlation was found with emotional functioning (*r* = 0.298, *p* = 0.002), while all other quality-of-life domains showed weaker positive associations (0.10 ≤ *r* ≤ 0.243).

### 3.4. Preferences for Information and Counseling Provision

Regarding preferences for information and counseling provision, 45.2% of participants (n = 56) rated the information received at the time of diagnosis as “very relevant,” as did 46.8% (n = 58) regarding the information provided throughout the treatment process. Concerning the timeliness of information, most participants (85.5%, n = 106) reported having received it “on time,” that is, at the moment when it was most needed or would have been most beneficial. Despite this, 37.1% of the participants (n = 46) expressed some dissatisfaction with how their additional informational needs were addressed. Among these, 30.4% (n = 14) specifically emphasized the need for more information on psychological support services. Others highlighted the importance of addressing topics such as nutrition, referral to physiotherapy, and the provision of information to support the preservation of sexual health. In terms of the most appropriate time to receive information about the disease and its side effects, 55.6% (n = 69) preferred a phased approach throughout the different stages of the intervention process. However, a substantial proportion (45.0%, n = 55) indicated a preference for receiving such information during the early stages of diagnosis. Lastly, 90.3% of participants (n = 112) considered face-to-face consultations with healthcare professionals to be the most appropriate method for addressing their informational needs. In contrast, only 4.8% (n = 6) regarded peer-based support—receiving information from someone with a similar oncological experience—as a suitable approach.

## 4. Discussion

The present study aimed to validate the SIPYF-CPS and to explore the informational needs of young adult female cancer survivors. The findings underscore the critical importance of addressing these needs from a multidimensional, developmentally sensitive perspective. Psychometric analyses revealed that the SIPYF-CPS, comprising 23 items originally but refined to a 22-item final version in this study, exhibits strong internal consistency—comparable to other established measures of supportive needs [29]—and construct validity, suggesting its relevance for assessing satisfaction with cancer-related information across four distinct domains: Disease-Related Information, Symptoms and Functional Limitations, Implications for Fertility and Parenthood, and Support Services. These results reinforce the scale’s conceptual foundation, based on Ormandy’s [13] theoretical framework, which highlights the contextual and temporal nature of informational needs.

This multidimensional structure contrasts with earlier conceptualizations, such as the preliminary unidimensional model proposed by Silva et al. [30], and aligns more closely with the recent international literature framing informational needs as complex and evolving constructs [14,15]. The identified domains also resonate with previous instruments, such as the SCNS-SF34 [29] and ISNQ [26], while expanding their scope by explicitly incorporating reproductive and fertility-related content—areas frequently overlooked in earlier tools [25].

Descriptive analysis of needs satisfaction revealed substantial dissatisfaction in key areas, particularly regarding fertility preservation, sexual health, alternative family-building strategies, and access to support services. Nearly half of the participants reported low satisfaction with information about the impact of cancer on their sexual life and fertility options—findings consistent with Benedict et al. [10,22], who identified similar gaps as major sources of distress and decisional conflict among young survivors. Variability in satisfaction related to fertility and parenthood may reflect unequal access to fertility care (e.g., only 22.6% of participants underwent fertility preservation) and inconsistencies in counseling practices across clinical settings, which may underlie informational disparities previously documented in the literature [3]. In the domain of support services, participants reported a lack of information about psychosocial support, peer groups, and complementary therapies. These results may reflect the “invisibility” of supportive services in cancer care systems, despite their crucial role in psychological adjustment, as recognized in the literature [45]. Faller et al. [46] further emphasized that poor communication about support resources significantly contributes to increased distress.

Importantly, the pattern of associations observed between informational satisfaction and psychosocial outcomes provides empirical support for the convergent validity of the SIPYF-CPS, as hypothesized. Consistent with theoretical expectations and the prior literature [33], lower satisfaction with information was moderately associated with higher symptoms of anxiety and depression (r = –0.464 to –0.500), reflecting medium-to-large effect sizes. These results underscore the clinical relevance of informational needs in shaping emotional vulnerability among young female cancer survivors. Similarly, as anticipated, moderate positive correlations were found between informational satisfaction and key quality-of-life domains—emotional, physical, and cognitive functioning (r = 0.309 to 0.397)—suggesting that access to timely and relevant information may serve as a protective factor against functional decline [34].

Correlations with social and role functioning were smaller but statistically significant, aligning with the hypothesized weaker associations for these domains and suggesting that other contextual variables may also be at play. Additionally, and in line with prior expectations, small but significant negative associations were found between reproductive concerns and satisfaction in the Disease-Related Information and Fertility and Parenthood domains (r = –0.294 to –0.320). While modest in magnitude, these associations reinforce the particular vulnerability of this population regarding fertility-related uncertainty and further validate the inclusion of fertility-specific content in the SIPYF-CPS. Taken together, these findings are consistent with the proposed hypotheses and support the scale’s sensitivity in capturing the multidimensional psychosocial implications of unmet informational needs.

Additionally, the study explored participants’ preferences regarding the timing and mode of information delivery, in line with Ormandy’s [13] framework, which suggests fluctuations in patients’ desire for and avoidance of information at different stages of the illness trajectory. In this study, most participants (55.6%) preferred a phased approach, receiving information gradually throughout diagnosis and treatment. However, a substantial portion (45.0%) favored early disclosure, especially regarding fertility and treatment side effects. These preferences highlight the need for dynamic, individualized communication strategies, consistent with Leydon et al. [47], who noted that patients’ informational needs and processing capacities shift over time. Equally noteworthy was the strong preference for face-to-face communication with healthcare professionals (reported by 90.3% of participants) compared to peer-based formats (preferred by only 4.8%). This underscores the importance of trust, empathy, and professional expertise in meeting informational needs—factors that digital or informal channels may not adequately provide. These findings challenge the growing emphasis on online and peer-delivered interventions, suggesting that while such tools can complement care, they cannot substitute for high-quality interpersonal communication in this population.

Despite its contributions, the present study has several limitations that should be considered when interpreting the findings. The use of a convenience sample recruited through online platforms may have introduced selection bias, favoring participants with higher digital access and health engagement. Additionally, the inclusion of a heterogeneous sample in terms of cancer types and treatment stages may have masked condition-specific informational needs. The cross-sectional design limits causal interpretations, particularly regarding the directionality of associations between informational satisfaction and psychosocial outcomes, as well as potential fluctuations in informational needs across the cancer trajectory. Furthermore, several key psychometric properties of the SIPYF-CPS—such as sensitivity to change, known-groups validity, and predictive validity—were not evaluated in this study, limiting the full assessment of the scale’s robustness.

Future research should adopt longitudinal designs to examine how informational needs evolve over time and influence clinical outcomes. Subgroup analyses are also warranted, particularly comparing women by cancer type, treatment history, or fertility status. Cross-cultural validation of the SIPYF-CPS would further enhance its utility, especially given the variability in informational expectations across contexts. Moreover, studies incorporating healthcare professionals’ perspectives could help identify mismatches between the information provided and patients’ perceived needs, paving the way for more effective training and system-level improvements.

## 5. Conclusions

In conclusion, this study provides important initial evidence supporting the reliability and clinical relevance of the SIPYF-CPS for assessing informational needs in young female cancer survivors. The findings underscore the value of early, ongoing, and personalized information delivery—particularly concerning fertility, sexual health, and support services. Addressing these unmet needs through structured communication and patient-centered care may substantially improve psychological well-being and long-term survivorship outcomes in this vulnerable population. Future research is needed to further validate and refine the instrument across diverse samples and settings.

## Figures and Tables

**Figure 1 healthcare-13-01757-f001:**
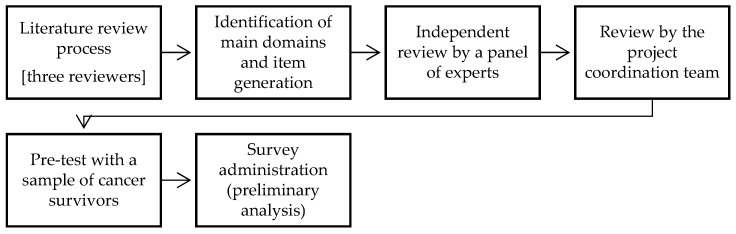
Development process of the SIPYF-CPS [based on Silva [31].

**Table 1 healthcare-13-01757-t001:** Sample description (n = 124).

Variable	n	%
Age in years (M ± SD; range)	38.18 ± 5.49, 21–45	
**Education**		
Primary school	1	0.8
Middle school	4	3.2
High school	38	30.6
University	81	65.3
**Marital status**		
Single	29	23.4
Married	53	42.7
Divorced/separated	10	8.7
Cohabiting	32	25.8
**Employment status**		
Employed/Self-employed	81	65.4
Unemployed	6	4.8
Student	1	0.8
**Variable**	**n**	**%**
Disability pension	3	2.4
Sick leave	30	24.2
Other	3	2.4
Age at diagnosis in years (M ± SD; range)	35.77 ± 5.67, 18–44	
**Cancer type**		
Breast	95	76.6
Hodgkin or Non-Hodgkin Lymphoma	11	8.8
Cervix	4	3.2
Ovary	5	4
Colorectal	2	1.6
Lung	3	2.4
Other (e.g., leukemia, stomach, bladder, multiple myeloma)	4	3.2
**Previous treatments**		
Surgery	80	64.5
Chemotherapy	100	80.6
Radiation therapy	67	54
Hormone therapy	48	38.7
Targeted therapy	13	10.5
Immunotherapy	35	28.2
Stem cell transplants	2	1.6
Other	2	1.6
**Current stage of the cancer** Treatment		
Undergoing treatment	81	65.3
Follow-up	43	34.7
**Other chronic conditions**		
None	92	74.2
Hypertension	3	2.4
Diabetes	2	1.6
Asthma	7	5.6
Other (e.g., kidney disease, cardiovascular disease, musculoskeletal disorders)	20	16.1
**Use of mental health services**		
Yes	57	46
No	67	54
**Motherhood**		
One or more biological children	73	58.9
No biological children	51	41.1
**Menstrual Cycle Status**		
Regular	21	16.9
Irregular	11	8.9
**Underwent fertility care before treatment**		
Yes	28	22.6
No	96	77.4
**Wants a (or another) biological child**		
Yes	41	33.1
No	83	66.9
**Location of medical care provision**		
Public hospital	90	72.6
Private hospital	18	14.5
Both	16	12.9

**Table 2 healthcare-13-01757-t002:** Descriptive analysis of the items: frequencies, medians (Mdn), and interquartile range (IQR) of the SIPYF-CPS (n = 124).

Item	Item Description	Mdn (IQR)	Not at All Satisfied n (%)	Slightly Satisfied n (%)	Moderately Satisfied n (%)	Quite Satisfied N (%)	Very Satisfied (%)
1	Cancer type	4 (2)	8 (6.5)	10 (8.1)	35 (28.2)	34 (27.4)	37 (29.8)
2	Cancer stage	3 (1)	11 (8.9)	16 (12.9)	43 (34.7)	27 (21.8)	27 (21.8)
3	Treatment options (e.g., discussion of available cancer treatment alternatives)	3 (1.75)	11 (8.9)	11 (8.9)	41 (33.1)	30 (24.2)	31 (25)
4	Expected physical symptoms during treatments	4 (1)	8 (6.5)	14 (11.3)	30 (24.2)	48 (38.7)	24 (19.4)
5	Expected emotional symptoms during treatments	3 (2)	21 (16.9)	21 (16.9)	43 (34.7)	20 (16.1)	19 (15.3)
6	Prognosis in terms of recovery	3.5 (1)	5 (4)	19 (15.3)	38 (30.6)	41 (33.1)	21 (16.9)
7	Risk of infertility	3 (1)	19 (15.3)	10 (8.1)	38 (30.6)	27 (21.8)	29 (23.4)
8	Risk of early menopause	3 (2)	20 (16.1)	15 (12.1)	36 (29)	27 (21.8)	26 (21)
9	Fertility preservation options	3 (2)	29 (23.4)	18 (14.5)	28 (22.6)	24 (19.4)	25 (20.2)
10	Physical side effects of treatments	3 (2)	13 (10.5)	33 (26.6)	38 (30.6)	26 (21)	14 (11.3)
11	Restrictions in daily activities during treatments	3 (2)	7 (5.6)	29 (23.4)	41 (33.1)	29 (23.4)	18 (14.5)
12	Work-related restrictions and/or return to work	3 (2)	16 (12.9)	24 (19.4)	49 (39.5)	17 (13.7)	18 (14.5)
13	Limitations in leisure activities and/or hobbies	3 (2)	8 (6.5)	27 (21.8)	47 (37.9)	23 (18.5)	19 (15.3)
14	Services provided by healthcare providers (e.g., psychological support, social services, gynecology)	3 (2)	21 (16.9)	26 (21)	39 (31.5)	20 (16.1)	18 (14.5)
15	Other organizations offering services to cancer patients (e.g., associations)	3 (2)	27 (21.8)	15 (12.1)	50 (40.3)	16 (12.9)	16 (12.9)
16	Existence of informal support groups	2 (2)	38 (30.6)	27 (21.8)	34 (27.4)	13 (10.5)	12 (9.7)
17	Complementary and alternative treatments (e.g., Reiki, acupuncture)	2 (2)	46 (37.1)	26 (21)	25 (20.2)	13 (10.5)	14 (11.3)
18	Surveillance recommendations (e.g., self-examination)	3 (2)	18 (14.5)	16 (12.9)	49 (39.5)	19 (15.3)	22 (17.7)
19	Risk behaviors	3 (2)	17 (13.7)	27 (21.8)	40 (32.3)	20 (16.1)	20 (16.1)
20	Restrictions in daily activities after recovery	3 (2)	14 (11.3)	21 (16.9)	49 (39.5)	20 (16.1)	20 (16.1)
21	Implications for sexual life	3 (3)	45 (36.3)	16 (12.9)	28 (22.6)	19 (15.3)	16 (12.9)
22	Genetic risk for offspring	3 (2)	22 (17.7)	11 (8.9)	32 (25.8)	29 (23.4)	30 (24.2)
23	Alternatives to family building/expansion (e.g., adoption)	3 (1.75)	31 (25)	27 (21.8)	37 (29.8)	13 (10.5)	16 (12.9)

**Table 3 healthcare-13-01757-t003:** Item properties: normality, item–total correlation, and reliability (n = 124).

Item	Min-Max	Skewness	Kurtosis	Item–Total Correlation	Cronbach’s Alpha if Item Deleted
1	1–5	−0.595	−0.371	0.616	0.941
2	1–5	−0.250	−0.711	0.655	0.941
3	1–5	−0.429	−0.561	0.695	0.940
4	1–5	−0.608	−0.263	0.644	0.941
5	1–5	0.029	−0.896	0.693	0.940
6	1–5	−0.296	−0.536	0.658	0.941
7	1–5	−0.359	−0.889	0.655	0.941
8	1–5	−0.238	−10.014	0.628	0.941
9	1–5	−0.037	−10.319	0.515	0.943
10	1–5	0.111	−0.787	0.786	0.939
11	1–5	0.031	−0.763	0.719	0.940
12	1–5	0.105	−0.655	0.662	0.941
13	1–5	0.093	−0.651	0.693	0.940
14	1–5	0.113	−0.930	0.678	0.940
15	1–5	0.060	−0.819	0.422	0.944
16	1–5	0.476	−0.781	0.528	0.942
17	1–5	0.614	−0.849	0.308	0.946
18	1–5	−0.070	−0.770	0.694	0.940
19	1–5	0.090	−0.902	0.717	0.940
20	1–5	0.000	−0.671	0.738	0.939
21	1–5	0.335	−1.242	0.689	0.940
22	1–5	−0.357	−1.053	0.619	0.941
23	1–5	0.356	−0.875	0.597	0.941

**Table 4 healthcare-13-01757-t004:** Model fit indices for exploratory factor solutions with one to five factors (n = 124).

Number of Factors	Χ^2^	Χ^2^/df	CFI	RMSEA (90% CI)	SRMR
**Original 23-item version**					
1	864.64 *	3.76	0.86	0.15 (0.14–0.16)	0.10
2	595.69 *	2.86	0.92	0.12 (0.11–0.13)	0.08
3	448.67 *	2.40	0.94	0.11 (0.09–0.12)	0.06
4	340.46 *	2.04	0.96	0.09 (0.08–0.11)	0.05
**22-Item Version (Modified)**					
4	279.56 *	1.88	0.97	0.08 (0.07–0.10)	0.04

* *p* < 0.001.

**Table 5 healthcare-13-01757-t005:** Standardized factor loadings, factor intercorrelations, and internal consistency coefficients (Cronbach’s α and McDonald’s ω) from the initial exploratory factor analysis with 23 items (n = 124).

Item	F1 Disease-Related Information	F2 Symptoms and Functional Limitations	F3 Implications for Fertility and Parenthood	F4 Support Services
1	**0.98**	−0.06	0.02	−0.08
2	**0.80**	−0.02	0.14	−0.03
3	**0.59**	0.13	0.06	0.20
4	0.21	**0.39**	0.25	0.08
5	0.13	**0.57**	0.07	0.20
6	**0.46**	0.28	0.14	0.00
7	0.05	−0.03	**0.93**	−0.01
8	0.12	−0.02	**0.75**	0.06
9	−0.13	0.13	**0.76**	0.00
10	0.20	**0.35**	0.31	0.19
11	−0.03	**0.86**	0.11	−0.02
12	−0.02	**0.87**	0.06	−0.07
13	−0.04	**0.99**	−0.03	−0.05
14	0.21	0.36	−0.02	**0.41**
15	−0.11	−0.01	0.00	**0.88**
16	0.00	−0.09	0.08	**0.92**
17	0.18	0.09	−0.20	**0.47**
18	**0.45**	0.19	0.06	0.28
19	**0.50**	0.30	−0.05	0.25
20	0.34	**0.54**	−0.02	0.12
21	0.25	**0.45**	0.10	0.17
22	0.26	0.22	0.20	0.19
23	0.09	0.25	**0.38**	0.15
Intercorrelations				
F1	1			
F2	0.56	1		
F3	0.54	0.60	1	
F4	0.40	0.38	0.37	1
Cronbach’s Alpha (α)	0.885	0.917	0.822	0.738
McDonald’s Omega (ω)	0.884	0.916	0.824	0.750

Note: The stronger factor loadings are indicated in bold font.

**Table 6 healthcare-13-01757-t006:** Descriptive statistics for the total SIPYF-CPS score and its dimensions (n = 124).

Variable	Min	Max	Mean	SD
SIPYF-CPS Total Score	28	110	67.07	18.45
F1—Disease-Related Information	1	5	3.33	0.96
F2—Symptoms and Functional Limitations	1.22	5	3.03	0.95
F3—Implications for Fertility and Parenthood	1	5	3.03	1.10
F4—Support Services	1	5	2.65	0.98

## Data Availability

The raw data supporting the conclusions of this article will be made available by the authors on request.

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
