# Peer review of "Mapping the Unmet Informational Needs of Young Portuguese Female Cancer Survivors: Psychometric Validation of a Multidimensional Scale"

_healthcare, 2025, doi:10.3390/healthcare13141757_

Round 1
Reviewer 1 Report
Comments and Suggestions for Authors
Dear authors, the article is an important subject. The writing flow of the article is confusing. There are serious technical, numerical and analysis errors in the statistics and tables.
Author Response
REVIEWER 1
Dear authors, the article is an important subject. The writing flow of the article is confusing. There are serious technical, numerical and analysis errors in the statistics and tables.
A: Thank you very much for your valuable feedback. We would like to clarify that the issues identified were only typographical errors, which have now been carefully corrected. There were no technical, numerical, or analytical errors in the statistics and tables. We appreciate your attention and have made the necessary adjustments to improve the clarity and flow of the manuscript.
Reviewer 2 Report
Comments and Suggestions for Authors
The scientific paper is well done, methodologically correct, the language is academic and complies with the validation aspects of a scale.
However, some comments:
1.- It seems an insufficient number for pretest phase only 3 patients. They should justify it better or indicate this limitation more clearly when discussing content validity
2.- Somewhere, probably even in the title, it should be mentioned that the scale is validated for Portuguese population and that it is in Portuguese language.
3.- Usually it is recommended to survey 10 people per item to validate, if your questionnaire has 23 items, it would have been necessary to survey at least 230 women, do you consider that your sample size has been adequate to adequately validate the tool presented?
Thank you very much
Author Response
REVIEWER 2
The scientific paper is well done, methodologically correct, the language is academic and complies with the validation aspects of a scale.
However, some comments:
1.- It seems an insufficient number for pretest phase only 3 patients. They should justify it better or indicate this limitation more clearly when discussing content validity.
A: We appreciate the reviewer’s observation regarding the small number of participants (n = 3) in the pretest phase aimed at ensuring content validity. We agree that this is a limited sample and have now made this limitation more explicit in the revised manuscript. However, our decision was based on methodological recommendations suggesting that small samples — typically between 3 and 5 participants — may be adequate for the initial pretesting of self-administered instruments, especially when the goal is to assess clarity, comprehension, and acceptability rather than to conduct statistical validation (e.g., Boateng et al., 2018). In addition, we asked these participants to provide open-ended feedback and also conducted a group discussion. This approach generated valuable qualitative insights based on both individual and collective perspectives, allowing for small but meaningful refinements that improved the final version’s clarity and appropriateness. To further strengthen content validity, a panel of psycho-oncology experts — with direct experience working with the target population — also reviewed the instrument to ensure its conceptual adequacy and clinical relevance. Furthermore, in a preliminary phase of psychometric evaluation, 39 participants were invited to suggest potential improvements to the measure, which provided additional insights into its clarity and relevance from the end-users’ perspective.
2.- Somewhere, probably even in the title, it should be mentioned that the scale is validated for Portuguese population and that it is in Portuguese language.
A: Thank you for your helpful suggestion. We agree with your observation and have revised the title accordingly to clearly indicate that the scale has been validated for the Portuguese population and that it is in the Portuguese language.
3.- Usually it is recommended to survey 10 people per item to validate, if your questionnaire has 23 items, it would have been necessary to survey at least 230 women, do you consider that your sample size has been adequate to adequately validate the tool presented?
A: Thank you for your insightful comment. We acknowledge that a larger sample size—commonly around 10 participants per item—is generally recommended to strengthen the validation of psychometric instruments. In our case, the sample size did not reach this ideal but did meet the minimum guideline of five participants per item, which is supported in the literature as acceptable for exploratory factor analysis (e.g., Bryant & Yarnold, 1995). We also note the particular challenges in recruiting a homogeneous and specific population such as young female cancer survivors, which limited the sample size achievable in this study. Despite these constraints, our sample allowed for initial validation and provided valuable insights. Nevertheless, we agree that future research with larger and more diverse samples is needed to further confirm the robustness and generalizability of the SIPYF-CPS.
Reviewer 3 Report
Comments and Suggestions for Authors
The present study aimed to: (i) explore the reliability and construct validity of the Satisfaction with Information Provided to Young Oncology Patients Scale (SIPYF-CPS); (ii) identify the main unmet needs of 124 reproductive-age female cancer survivors; and (iii) characterize their preferences regarding the delivery of information and counselling. The methodological approach adopted in the psychometric sub-study lies within the framework of classical test theory, one of the usual approaches for development or psychometric study of measurement instruments. The results obtained seem to support the view that SIPYF-CPS is a valid and reliable tool for assessing informational needs in young female cancer survivors. Although the topic of the manuscript is of potential interest, some parts of the study/manuscript need further clarification or should be significantly reworked. More detailed comments/suggestions, structured according to the sections of the paper, follow.
INTRODUCTION
[Objectives]
1.- Authors should include hypotheses regarding validity evidence based on relationships with other variables (convergent validity in authors’ terms) in the last paragraph of the introduction. The COnsensus-based for the Selection of health Measurement INstruments (COSMIN) approach recommends constructing hypothesis for relative correlation sizes of the different comparator instruments (e.g., the correlation of the instrument/subscale of interest with instrument A is expected to be higher than its correlation with instrument B). In any case, it would be desirable to also construct hypotheses for the absolute magnitude of some particular correlations.
MATERIALS AND METHODS
[Data analysis]
2.- I wonder why the authors have not opted for a confirmatory factor analysis (CFA) approach to do a cross-validation of the theoretical SIPYF-CPS four-dimension structure (i.e., the four key content domains of the SIPYF-CPS; page 4, lines 171-185).
3.- In any case, the EFA suffers from several serious limitations that need to be addressed. Of main concern is the use of the “Little-Jiffy” approach (Kaiser & Rice, 1974), which involves the use of principal component analysis (PCA) for factor extraction, Kaiser’s eigenvalues-greater-than-one rule for factor-retention, and orthogonal Varimax for factor rotation. These methods have many drawbacks (e.g., i) PCA is only a data reduction method and not, strictly speaking an exploratory factor analysis (EFA) approach; ii) parallel analysis, MAP, analysis of residual correlations, model fit indices are best suited procedures for deciding the number of factors to retain; iii) in the psychological and biomedical sciences we generally expect some correlation among factors, since behaviour is rarely partitioned into neatly packaged units that function independently of one another. Therefore, using orthogonal rotations potentially results in a less useful solution where factors are correlated) and for this reason several researchers have strongly discouraged their use (e.g., Costello & Osborne, 2005; Fabrigar & Wegener, 2012; Lloret-Segura et al., 2014). Automatically applying popular (and overused) approaches and/or default options present in mainstream statistical software (e.g., IBM SPSS) can lead to inappropriate or erroneous decisions that compromise the integrity of the EFA results. We, therefore, respectfully but emphatically urge the authors to not use the “Little-Jiffy” approach.
RESULTS
[Sociodemographic and clinical characteristics of the sample]
4.- It is unlikely that all approached potential participants consented to participate. It would be of interest to know if those patients who refused to participate, compared with those who accepted, differed in terms of demographic or clinical characteristics.
[Dimensionality – Factor analysis]
5.- Please see above (comment #3).
[Dimensionality – Factor analysis]
6.- As stated by the authors themselves (page 3, lines 137-138), “The present study aimed to: (i) explore the psychometric properties (reliability and construct validity) of SIPYF-CPS […]”. Therefore, the item pruning process –or the development of a short version of the original scale– performed [cf. page 12, lines 345-349: “During this process, it was found that items 4 ("expected physical symptoms during treatment"), 14 ("services offered by the healthcare provider unit"), 22 ("genetic risk for offspring"), and 23 ("alternatives for family construction/continuation") had factor loadings below 0.50. These items were therefore excluded, and the PCA was rerun using the remaining 19 items”] is unexpected. This point needs further clarification. Moreover, the rationale for eliminating items with component loadings below 0.50 should be explained and referenced.
DISCUSSION
7.- When discussing convergent validity results, the magnitude or the effect size of the findings should be discussed when interpreting the results.
[Limitations paragraph]
8.- Several metric characteristics of the SIPYF-CPS were not assessed in the present study (e.g., sensitiveness to change, known-groups validity, predictive validity). This fact should be specifically acknowledged as additional limitations of the study.
[Conclusion paragraph]
9.- Consider tempering/rewording the sentence ‘In conclusion, this study validates the SIPYF-CPS as a […]’ (page 16, line 496). Any validation process is an ongoing process –even a never ending one according to some authors– which requires new assessments to acquire sufficient evidence in different samples and contexts. Neither reliability nor validity are immutable or inherent properties of a measurement instrument (they result from the interaction among the scale, the specific sample assessed, and the assessment situation).
In a similar way, the terms ‘validation’ and ‘validate’ should be replaced in the title and abstract, as well as throughout the manuscript.
10- Overall
The quality of language in the manuscript is generally acceptable. However, the text needs some typos corrections to be made (e.g., M±DP [pp. 7-8, Table 1 --> SD]; M±DP [p. 13, Table 4 --> SD]; ESIDOJ-23 [p. 17, line 22 --> This is a 19-item measure]).
Review references
Costello, A. B., & Osborne, J. W. (2005). Best practices in exploratory factor analysis: Four 664 recommendations for getting the most from your analysis. Practical Assessment, Research & Evaluation, 10, 7. https://doi.org/10.7275/jyj1-4868
Fabrigar, L. R., & Wegener, D. T. (2012). Understanding statistics: Exploratory factor analysis. New York, NY: Oxford University.
Kaiser, H. F., & Rice, J. (1974). Little Jiffy, Mark IV. Educational and Psychological Measurement, 34(1), 111–117. https://doi.org/10.1177/001316447403400115
Lloret-Segura, S., Ferreres-Traver, A., Hernández-Baeza, A., & Tomás-Marco, I. (2014). Exploratory item factor analysis: A practical guide revised and updated. Anales de Psicología / Annals of Psychology, 30(3), 1151–1169. https://doi.org/10.6018/analesps.30.3.199361
Author Response
REVIEWER 3
The present study aimed to: (i) explore the reliability and construct validity of the Satisfaction with Information Provided to Young Oncology Patients Scale (SIPYF-CPS); (ii) identify the main unmet needs of 124 reproductive-age female cancer survivors; and (iii) characterize their preferences regarding the delivery of information and counselling. The methodological approach adopted in the psychometric sub-study lies within the framework of classical test theory, one of the usual approaches for development or psychometric study of measurement instruments. The results obtained seem to support the view that SIPYF-CPS is a valid and reliable tool for assessing informational needs in young female cancer survivors. Although the topic of the manuscript is of potential interest, some parts of the study/manuscript need further clarification or should be significantly reworked. More detailed comments/suggestions, structured according to the sections of the paper, follow.
INTRODUCTION
[Objectives]
1.- Authors should include hypotheses regarding validity evidence based on relationships with other variables (convergent validity in authors’ terms) in the last paragraph of the introduction. The COnsensus-based for the Selection of health Measurement INstruments (COSMIN) approach recommends constructing hypothesis for relative correlation sizes of the different comparator instruments (e.g., the correlation of the instrument/subscale of interest with instrument A is expected to be higher than its correlation with instrument B). In any case, it would be desirable to also construct hypotheses for the absolute magnitude of some particular correlations.
A: We appreciate the reviewer’s valuable suggestion regarding the inclusion of hypotheses for construct validity based on relationships with other variables. In line with COSMIN recommendations, we have revised the final paragraph of the introduction (Section 1.2. Objectives and Hypotheses) to include theory-driven hypotheses concerning expected associations between the SIPYF-CPS and related constructs.
Specifically, we now outline that:
- Higher satisfaction with informational needs (SIPYF-CPS) is expected to be associated with lower levels of psychological distress (i.e., symptoms of anxiety and depression).
- Lower satisfaction is also anticipated to be linked to higher reproductive concerns, which are known to be particularly relevant for young female cancer survivors.
- In addition, positive associations are expected between SIPYF-CPS scores and quality of life domains—particularly emotional, cognitive, and physical functioning—whereas weaker associations are anticipated with social and role functioning.
These hypotheses aim to guide the assessment of convergent validity, based on established literature and conceptual relevance between constructs.
MATERIALS AND METHODS
[Data analysis]
2.- I wonder why the authors have not opted for a confirmatory factor analysis (CFA) approach to do a cross-validation of the theoretical SIPYF-CPS four-dimension structure (i.e., the four key content domains of the SIPYF-CPS; page 4, lines 171-185).
A: We sincerely appreciate the reviewer’s thoughtful and insightful comments regarding our factor analytic approach. We fully acknowledge the importance of cross-validating the theoretical structure of the SIPYF-CPS through confirmatory factor analysis (CFA). However, as this study represents the initial validation of a newly developed instrument, our primary aim was to explore the latent structure of the scale within a development sample using exploratory factor analysis (EFA). As recommended in the psychometric literature (e.g., Boateng et al., 2018), EFA is typically the first step when the dimensionality of a scale has not yet been empirically established. We agree that a subsequent and essential step will involve applying CFA to test the hypothesized four-factor model on an independent sample, thereby providing a more rigorous assessment of the scale’s factorial validity.
3.- In any case, the EFA suffers from several serious limitations that need to be addressed. Of main concern is the use of the “Little-Jiffy” approach (Kaiser & Rice, 1974), which involves the use of principal component analysis (PCA) for factor extraction, Kaiser’s eigenvalues-greater-than-one rule for factor-retention, and orthogonal Varimax for factor rotation. These methods have many drawbacks (e.g., i) PCA is only a data reduction method and not, strictly speaking an exploratory factor analysis (EFA) approach; ii) parallel analysis, MAP, analysis of residual correlations, model fit indices are best suited procedures for deciding the number of factors to retain; iii) in the psychological and biomedical sciences we generally expect some correlation among factors, since behaviour is rarely partitioned into neatly packaged units that function independently of one another. Therefore, using orthogonal rotations potentially results in a less useful solution where factors are correlated) and for this reason several researchers have strongly discouraged their use (e.g., Costello & Osborne, 2005; Fabrigar & Wegener, 2012; Lloret-Segura et al., 2014). Automatically applying popular (and overused) approaches and/or default options present in mainstream statistical software (e.g., IBM SPSS) can lead to inappropriate or erroneous decisions that compromise the integrity of the EFA results. We, therefore, respectfully but emphatically urge the authors to not use the “Little-Jiffy” approach.
A: We thank the reviewer for the thorough and constructive critique regarding our initial use of the “Little-Jiffy approach” (i.e., PCA extraction, Kaiser’s eigenvalue >1 rule, and Varimax rotation). We acknowledge that this approach has well-documented limitations, especially in psychological measurement contexts, as highlighted by Costello & Osborne (2005) and Fabrigar & Wegener (2012). Considering these concerns, we completely re-conducted our exploratory factor analysis (EFA) using a more appropriate and rigorous methodology. Specifically, we applied the WLSMV estimator, which is suitable for ordinal data, along with oblique (oblimin) rotation to allow for correlated factors. Furthermore, instead of relying on Kaiser’s eigenvalue >1 criterion, we based factor retention decisions on multiple model fit indices (CFI, RMSEA, SRMR), theoretical interpretability, and the pattern of factor loadings. This revised analysis produced a more robust, interpretable, and psychometrically sound four-factor solution, as presented in the updated results (Tables 4 and 5). We believe this approach adequately addresses the reviewer’s concerns and strengthens the validity of the factor structure reported.
RESULTS
[Sociodemographic and clinical characteristics of the sample]
4.- It is unlikely that all approached potential participants consented to participate. It would be of interest to know if those patients who refused to participate, compared with those who accepted, differed in terms of demographic or clinical characteristics.
A: We appreciate the reviewer’s insightful comment. However, it is not possible for us to compare the demographic or clinical characteristics of participants who declined to participate with those who accepted, as data collection was conducted online and anonymously. Most individuals who did not proceed with the questionnaire exited the survey before providing any sociodemographic information. As such, we do not have data available to perform this comparison.
5.- Please see above (comment #3).
[Dimensionality – Factor analysis]
6.- As stated by the authors themselves (page 3, lines 137-138), “The present study aimed to: (i) explore the psychometric properties (reliability and construct validity) of SIPYF-CPS […]”. Therefore, the item pruning process –or the development of a short version of the original scale– performed [cf. page 12, lines 345-349: “During this process, it was found that items 4 ("expected physical symptoms during treatment"), 14 ("services offered by the healthcare provider unit"), 22 ("genetic risk for offspring"), and 23 ("alternatives for family construction/continuation") had factor loadings below 0.50. These items were therefore excluded, and the PCA was rerun using the remaining 19 items”] is unexpected. This point needs further clarification. Moreover, the rationale for eliminating items with component loadings below 0.50 should be explained and referenced.
A: Initially, during the first analysis using the “Little-Jiffy” approach (PCA), we excluded items with component loadings below 0.50, following common but sometimes overly strict conventions. Recognizing the limitations of PCA and that such rigid cutoffs may be inappropriate, our revised EFA adopted a more flexible and theory-driven approach. We retained items with loadings ≥ 0.40 when they showed good theoretical relevance and acceptable psychometric properties. The only item excluded in the revised analysis was item 22 (“genetic risk for offspring”), due to consistently low loadings (< 0.40) and high residual variance, indicating poor fit. We have clarified these criteria in the manuscript and referenced appropriate guidelines (e.g., Hair et al., 2014; Tabachnick & Fidell, 2013) supporting the use of 0.40 as a minimum threshold in similar contexts.
References:
- Hair, J. F., Black, W. C., Babin, B. J., & Anderson, R. E. (2014). Multivariate data analysis (7th ed.). Pearson Education.
- Tabachnick, B. G., & Fidell, L. S. (2013). Using multivariate statistics (6th ed.). Pearson Education.
DISCUSSION
7.- When discussing convergent validity results, the magnitude or the effect size of the findings should be discussed when interpreting the results.
A: Thank you for your valuable comment. We have revised the discussion section to explicitly address the magnitude of the correlations observed in the convergent validity analyses. The interpretation now clearly reflects the effect sizes in terms of their practical significance, distinguishing between small, moderate, and stronger associations, and contextualizing these findings within the relevant literature. We believe this addition strengthens the interpretation of the results and provides a more nuanced understanding of the SIPYF-CPS’s convergent validity.
[Limitations paragraph]
8.- Several metric characteristics of the SIPYF-CPS were not assessed in the present study (e.g., sensitiveness to change, known-groups validity, predictive validity). This fact should be specifically acknowledged as additional limitations of the study.
A: Thank you for your insightful comment. We have now explicitly acknowledged in the discussion section that several important psychometric properties of the SIPYF-CPS—such as sensitivity to change, known-groups validity, and predictive validity—were not evaluated in the present study. We recognize these as important limitations and highlight the need for future research to address these gaps in order to fully establish the scale’s applicability and robustness in both clinical and research contexts.
[Conclusion paragraph]
9.- Consider tempering/rewording the sentence ‘In conclusion, this study validates the SIPYF-CPS as a […]’ (page 16, line 496). Any validation process is an ongoing process –even a never ending one according to some authors– which requires new assessments to acquire sufficient evidence in different samples and contexts. Neither reliability nor validity are immutable or inherent properties of a measurement instrument (they result from the interaction among the scale, the specific sample assessed, and the assessment situation).
A: Thank you for your comment. We agree that validation is an ongoing process and that reliability and validity depend on the sample and context. We have revised the conclusion to clarify that this study provides initial evidence supporting the SIPYF-CPS, while further validation is needed in other samples and settings.
10- Overall
The quality of language in the manuscript is generally acceptable. However, the text needs some typos corrections to be made (e.g., M±DP [pp. 7-8, Table 1 --> SD]; M±DP [p. 13, Table 4 --> SD]; ESIDOJ-23 [p. 17, line 22 --> This is a 19-item measure]).
A: Thank you for pointing out these typographical errors. We have carefully reviewed the manuscript and corrected all identified mistakes, including the notation of standard deviation (SD) and the accurate description of the measure.
Round 2
Reviewer 3 Report
Comments and Suggestions for Authors
In my opinion, the quality of the manuscript has considerably improved and further review on my part is unnecessary. The authors have satisfactorily addressed most of my comments and concerns raised on their original submission.
In any case, I leave it to the editor to take a decision on whether the authors’ response is satisfactory.